**Data Availability Statement:** Data cannot be shared publicly due to ethical restrictions. Due to the inclusion of sensitive and potentially identifiable

# Screening to understand pregnancy preferences and offer referrals and treatment (SUPPORT): Results of a pilot quality improvement initiative

**Elizabeth Janiak**[1,2,3]*, **Kathryn Rexrode**[2,4], **Leah Santacroce**[5], **Sarah L. Johns**[1,3], **Maya Behn**[2,4], **Kari P. Braaten**[1,2,4], **Candace H. Feldman**[2,5]

1 Department of Obstetrics and Gynecology, Brigham and Women's Hospital, Boston, Massachusetts, United States of America, 2 Harvard Medical School, Boston, Massachusetts, United States of America, 3 Harvard TH Chan School of Public Health, Boston, Massachusetts, United States of America, 4 Department of Medicine, Division of Women's Health, Brigham and Women's Hospital, Boston, Massachusetts, United States of America, 5 Department of Medicine, Division of Rheumatology, Inflammation, and Immunity, Brigham and Women's Hospital, Boston, Massachusetts, United States of America

* ejaniak@bwh.harvard.edu

## Abstract

### Objective

To assess the feasibility of integrating a pregnancy intention assessment screening algorithm into the electronic medical record (EMR) at a multispecialty clinic focused on the health of women and people assigned female at birth (AFAB).

### Study design

This pilot quality improvement project implemented a series of clinician reminders, new data fields in the patient record, and templated clinical notes to prompt care providers across specialties to ask AFAB reproductive age individuals about their desire for future pregnancies. Investigators created a novel screening question based on prior literature and expert input. Prospective observational study of one year of during-intervention EMR data on screening uptake and documentation, contraceptive use, and referrals to obstetrics and gynecology (OBGYN) for preconception care, contraceptive care, and related services.

### Results

SUPPORT launched in February 2020 and was paused for 6 months due to the COVID-19 pandemic. During the intervention period through July 2021, 18% of patients for whom the automated screening reminder was activated had a documented pregnancy intention. Patients were screened in OBGYN, internal medicine, and eight subspecialty medical clinics. Among those screened, individuals who reported they did not desire pregnancy in the next year were more likely to use contraception (aOR 1.8, 95% CI 1.1, 3.1). Individuals that

data, the Massachusetts General Brigham Institutional Review Board approved sharing of study data only among members of the study team. Interested researchers can request access to the data by contacting the Board at partnersirb@partners.org.

**Funding:** This research was supported by internal funding provided through the Division of Women's Health at Brigham and Women's Hospital.The funders had no role in study design, data collection and analysis, decision to publish, or preparation of the manuscript.

**Competing interests:** The authors have declared that no competing interests exist.

did desire pregnancy in the next year were more likely to be subsequently referred to OBGYN (aOR 2.7, 95% CI 1.2, 6.0).

## Conclusions

Despite the competing demands of the COVID-19 pandemic, the SUPPORT intervention was utilized at higher rates than prior similar interventions and across multiple disease specialties.

## Implications

Results from the SUPPORT pilot suggest that pregnancy intention screening of reproductive age AFAB individuals with an EMR-based screening prompt is feasible at scale.

## Introduction

Pregnancy planning can help all people optimize preconception health, and particularly for individuals with chronic disease, help to prevent disease exacerbation, pregnancy complications, and fetal exposure to teratogenic medications with resulting developmental impacts [1]. However, female individuals with several chronic diseases, including sickle cell disease, epilepsy, and systemic rheumatic diseases, have been shown to receive contraceptive counseling and use contraception at lower rates than the general population [2–4]. Medical specialist and primary care providers who have robust longitudinal relationships with these high-risk patients enjoy an opportunity to improve maternal and infant health outcomes by asking patients about their conception plans and integrating appropriate, timely referrals to contraceptive or preconception services into their routine practice [5].

Prior research has yielded mixed results regarding the feasibility and impact of pregnancy intention assessment in both family planning and primary care settings. Several standardized tools for pregnancy intention assessment have been described in the literature. The proprietary One Key Question™ (OKQ) screener queries patients regarding their desire to become pregnant in the next year. The more complex PATH questions (Pregnancy Attitudes, Timing and How important is pregnancy prevention) explore multiple dimensions of pregnancy attitudes, including the strength of motivation to avoid pregnancy. OKQ has shown mixed effects on patient satisfaction, low uptake in an EMR-based primary care intervention and was associated with a decrease in documented pregnancy intention screening in a recent pilot study of veterans' healthcare [6–9]. A recent pilot study by our group found that in an intervention among rheumatologists, 9% of eligible patients were screened with OKQ and screening responses were correlated with contraceptive documentation and referrals; however, that pilot did not include the full OKQ training for providers [10]. A large cohort study included a contraceptive preferences survey as a research procedure and not a standard component of the clinical encounter and found that the PATH questions predicted contraceptive method choice, with those expressing a desire to wait to become pregnant and less happiness at the idea of pregnancy now being more likely to choose the most effective contraceptive methods [11].

Few prior studies have examined integration of pregnancy intention assessment into the care of people with chronic diseases. In a 2010 randomized trial comparing two approaches to clinical decision support (CDS) in the EMR to encourage prescription of contraception at the time a teratogenic medication is prescribed, both arms increased contraceptive documentation

slightly over baseline, but the change was not statistically significant [12]. Additional recent research on the integration of a comprehensive pregnancy intention assessment screening process into medical subspecialty care serving people with complex and chronic medical conditions is lacking.

To help fill this gap, we created and piloted Screening to Understand Pregnancy Preferences and Offer Referrals to Treatment (SUPPORT), a novel multi-component EMR-based intervention to prompt pregnancy intention assessment and facilitate appropriate clinical follow-up, documentation, and referral-making among primary care and specialty clinicians. We chose to design an original question that is customized to the cadence of an annual screening but that is also open-ended enough to accommodate preconception planning for people who desire pregnancy in the short-term (within a year) and for those who desire pregnancy in the longer term (in more than a year).

## Methods

### Setting

SUPPORT was created in cooperation with and piloted at Fish Center for Women's Health, a multispecialty ambulatory practice affiliated with Brigham and Women's Hospital in Boston, Massachusetts. The Fish Center provides clinical care focused on the needs of women and people assigned female at birth (AFAB) across the life course. The Fish Center houses a wide variety of clinical specializations and subspecialty clinics geared to specific populations, including cardiology, endocrinology, gynecology, hematology, mental health, nephrology, neurology, nutrition, pulmonology, rheumatology, sleep medicine, and several interdisciplinary clinical services such as a menopause and midlife clinic and a program to manage polycystic ovarian syndrome. While the primary care practices at Fish care for a wide variety of patients with and without chronic diseases, the co-location of many disease subspecialties in the clinic results in an overrepresentation of people with chronic conditions in the overall Fish patient population. Two investigators (KR and KPB) hold senior clinical leadership positions at the Fish Center and served as liaisons to clinical and administrative staff stakeholders throughout the development and execution of the intervention.

### Intervention design

We created a novel pregnancy intention assessment screener and follow-up algorithm with input from a multispecialty team including experts in trans-inclusive and trauma-informed care. As depicted in Fig 1, our screening question asks, "Do you think you would like to become pregnant in the future?" with the possible response options, "Yes, within a year"; "Yes, in more than a year"; "No"; "I'm not sure", and "I cannot become pregnant." This question and response options were integrated as a sheet into the medical history section of the Epic patient chart. A Best Practice Alert (BPA) flag was programmed to fire to remind the provider to fill in this sheet for all AFAB patients ages 18–50 who did not have a documented hysterectomy or other sterilizing procedure at baseline. In turn, once completed, the response chosen within the sheet would prompt additional BPAs to appear with customized decision support to encourage appropriate clinical follow-up, documentation and, as necessary, referrals for contraceptive, preconception, or fertility care (Fig 1). SUPPORT also included a templated text note for detailed documentation of the pregnancy intention assessment conversation and related decision support. SUPPORT is designed to screen each patient at least once annually; thus, the BPAs fire at every clinical visit until the screening question is completed, and then once annually.

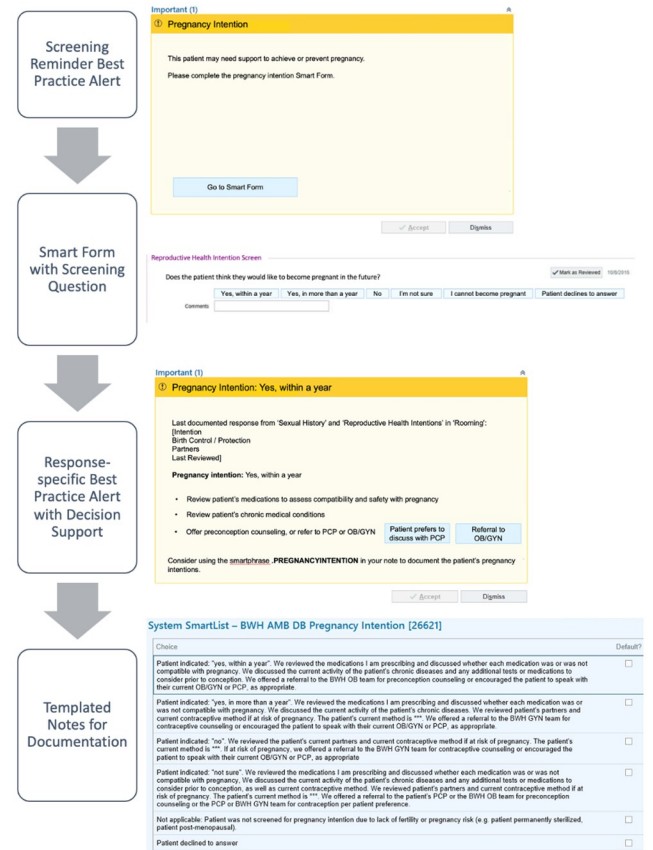

**Fig 1. SUPPORT Pilot EMR infrastructure and flow.** Diagram of intervention clinical flow, with screenshots of custom EMR build.

SUPPORT was rolled out via a series of pre-launch meetings with clinicians and staff at the Fish Center. All providers whose practices would incorporate the intervention were provided with a brief (<5 minutes) video training to watch at their convenience, and a written screening algorithm was posted in all exam rooms and clinical workspaces in the intervention practices to provide further reminders to use the SUPPORT infrastructure at the point of care. We offered on-site coaching on the day the new screening infrastructure was activated in the EMR, and ongoing opportunities for feedback to the study team.

## Study design

This prospective observational pilot study assessed the feasibility and utility of use of the SUPPORT infrastructure. We considered the following as indicators of feasibility: frequency of completion of the screening question and frequency of use of the documentation infrastructure. We considered the following as indicators of utility: likelihood of contraceptive documentation and of referrals in accordance with the screening answer choice. Contraceptive documentation was culled from medication lists and defined fields in the health or social history forms. Referrals were formal referrals in the EMR. We did not review free-text notes for either outcome. All data, including patient identifiers, were extracted from the EMR via the Research Patient Data Registry (RPDR) and the Epic Data Warehouse (EDW) at Mass General Brigham. Data were extracted on November 8, 2021. This study was approved by the Mass

General Brigham Institutional Review Board, which waived signed consent requirements for clinical staff who participated in the intervention and in pre- and post-intervention surveys, and waived patient consent for EMR review.

## Analysis

We ran chi-squared and t-tests to determine differences in demographic and healthcare utilization between eligible patients for whom screening was completed and those for whom it was not. We further explored the association between patient age group and screening response by conducting multinomial regression with pregnancy intention response as the outcome and age category as the primary predictor of interest, collapsing age 18–25 and age 35–50 to compare against age 26–35. Among individuals who were screened, we used logistic regression to determine the odds of contraceptive use on or within 90 days after screening fired and odds of referral on or within 90 days after screening, stratified by pregnancy intention response and adjusting for linear age. All tests were conducted with an alpha level of 0.05 and all analysis were conducted in SAS Version 9.4 and R version 4.1.2.

## Results

SUPPORT was formally launched in February 2020. In March 2020, we paused the project due to the increased demands on the healthcare system during the COVID-19 pandemic and the changes to EMR use that resulted from remote care. The SUPPORT infrastructure was reactivated in August 2020. For all analyses, we then defined the pilot intervention period as February 1st 2020 to March 1st 2020 and August 2nd 2020 to February 26th 2021 combined.

During the intervention period, the SUPPORT screening algorithm was activated for 79 clinicians during visits for 4004 AFAB individuals ages 18–50. Providers completed the pregnancy intention assessment question for 730 individuals (18.2%) and for 16 (2.2%) used the templated detailed notes to further document the screening conversation. Screened individuals were slightly younger compared to those who were not screened (Table 1, mean age 33.5 vs. 35.8, p<0.001). The racial distribution also differed between unscreened and screened individuals, with more screened individuals reporting Asian race, and more unscreened individuals reporting Black or African American race. Unscreened patients also had a higher average number of clinical visits during the study period. Additionally, more screened individuals attended office visits (80.1%) rather than telehealth, compared to unscreened individuals (63.0% office visits, p<0.001).

The greatest number of screenings were performed by OB/GYN (353 screenings) and internal medicine (258 screenings) clinicians. As a percentage of eligible encounters, screening was most frequent in OBGYN (27.0%), followed by hematology (21.1%) and endocrinology (18.2%) (Table 2). Half of individuals screened reported they do not want to become pregnant in the future, 22.0% would like to become pregnant more than a year into the future, and 9.7% were interested in pregnancy in the next year. Only 8.4% of individuals reported being unsure of their pregnancy intention. In a multinomial model comparing to those 25 and under or 35 and above combined, individuals 26–34 were more likely to be planning a future pregnancy within a year (OR 3.19 95% CI 1.89–5.38), in more than a year (OR 5.39 95% CI 3.61–8.04) or to be currently pregnant (OR 3.94 95% CI 2.27–6.86) compared to younger and older individuals.

Among 730 individuals screened, 343 (46.8%) had a contraceptive method documented on the date of service or within 90 days after screening, and 49 (6.7%) had a referral for contraception or preconception care on the date of service or within 90 days after screening. In an analysis excluding those who cannot become pregnant or were currently pregnant (n = 656) and accounting for age as a linear variable, individuals had higher odds of documented use of

**Table 1. SUPPORT pilot patient demographics.**

| Variable | Patients Screened (N = 730) | Eligible Patients Not Screened (N = 3,274) | P-Value |
|---|---|---|---|
| Age (mean, SD) | 33.5 (7.9) | 35.8 (8.5) | **<0.001** |
| Age Category (n, %) | | | **<0.001** |
| 18–25 | 118 (16.2) | 444 (13.6) | |
| 26–34 | 279 (38.2) | 946 (28.9) | |
| 35–50 | 333 (45.6) | 1,880 (57.5) | |
| Race (n, %) | | | **0.002** |
| White | 537 (73.6) | 2,361 (72.1) | |
| Asian | 75 (10.3) | 231 (7.1) | |
| Black or African American | 62 (8.5) | 408 (12.5) | |
| Other | 34 (4.7) | 155 (4.7) | |
| Declined/Unavailable | 22 (3.0) | 119 (3.6) | |
| Ethnicity (n, %) | | | 0.5894 |
| Hispanic | 53 (7.3) | 260 (7.9) | |
| Not Hispanic | 602 (82.5) | 2,646 (80.8) | |
| Declined/Unavailable | 75 (10.3) | 368 (11.2) | |
| Number of Outpatient Visits per individual (mean, SD) | 1.6 (1.0) | 1.8 (1.3) | **0.001** |
| Encounter Type (n, %) | | | **<0.001** |
| Office Visit | 585 (80.1) | 2,061 (63.0) | |
| Telemedicine | 145 (19.9) | 1,213 (37.0) | |

**Table 2. Provider specialty for patients screened during the SUPPORT Pilot[a].**

| Specialty Department | Screened Patients (N = 730) | Eligible Encounters (N = 5,791) | Percent of Eligible Encounters Screened |
|---|---|---|---|
| Cardiology | 1 | 52 | 1.9 |
| Dermatology | 51 | 1,749 | 3.4 |
| Endocrinology | 56 | 203 | 27.6 |
| Hematology | 4 | 16 | 25.0 |
| Neurology | 3 | 77 | 3.9 |
| OBGYN | 353 | 906 | 39.0 |
| Psychiatry | 1 | 25 | 4.0 |
| Rheumatology | 3 | 77 | 4.0 |
| Internal Medicine | 258 | 2,686 | 9.6 |

[a]The SUPPORT Pilot quality improvement project was conducted at a multispecialty clinic in Boston, MA from Feb 2020 through July 2021.

contraception on or within 90 days of the date of screening if they answered "I'm not sure" (aOR 2.51, 95% CI 1.24, 5.10), "Yes, in more than a year" (aOR 2.33, 95% CI 1.29, 4.21.), or "No" (aOR 1.80, 95% CI 1.06, 3.06), compared to "Yes, within a year" (reference group). Those who responded "Yes, within a year" were more likely to have a referral to obstetrics and gynecology within 90 days of screening, compared to those who responded "No" (aOR 2.69, 95% CI 1.20, 6.02), accounting for age (Table 3).

## Discussion

SUPPORT, a novel EMR-based pregnancy intention assessment and decision support algorithm, was used in nearly 1 in 5 clinical encounters with eligible patients, even in the midst of a

**Table 3. Logistic regression results for odds of contraception documentation and referral receipt among patients screened in the SUPPORT Pilot.**

| Estimates for screened individuals (N = 656)[1] | | |
|---|---|---|
| **Model** | **Adjusted Odds Ratio Estimate** | **95% Confidence Interval** |
| Contraceptive Use on or 90 days after BPA firing | | |
| Response | | |
| "Yes, within a year" (reference) | - | - |
| "Yes, in more than a year" | 2.33 | (1.29, 4.21) |
| "No" | 1.80 | (1.06, 3.06) |
| "I'm not sure" | 2.51 | (1.24, 5.10) |
| Referral on or 90 days after BPA firing | | |
| Response | | |
| "No" (reference) | - | - |
| "Yes, within a year" | 2.69 | (1.20, 6.02) |
| "Yes, in more than a year" | 1.91 | (0.57, 2.48) |
| "I'm not sure" | 0.49 | (0.11, 2.15) |

[a]The SUPPORT Pilot quality improvement project was conducted at a multispecialty clinic in Boston, MA from Feb 2020 through July 2021.

[1]All models removed individuals who responded "patient cannot become pregnant" and "patient is not currently pregnant". All models are adjusted for patient linear age in years.

global pandemic that created unprecedented strain on the US health care system. Screening was performed in eight medical specialties in addition to internal medicine and gynecology visits. Screening question responses were correlated with increased odds of contraceptive use and with increased odds of subsequent referral to obstetrics and gynecology. SUPPORT appears to be a feasible and useful intervention. However, use of the templated documentation infrastructure was very low (2% of encounters), suggesting that future interventions could exclude this component of the EMR infrastructure, or could better support clinicians in understanding why using a standardized note is beneficial (for example for data collection or cross-specialty collaboration) in comparison to other modes of documentation with which they are more familiar.

While the overall screening rate of 18% in the SUPPORT pilot trial was higher than the overall screening rates of 3–9% in recent studies of other standardized pregnancy intention assessment tools, our screening rate in internal medicine specifically was 9.6% [8, 9]. While the SUPPORT screening was completed at least once in eight medical subspecialties, some specialties had much higher rates of screening compared to others. The reasons for these differences are not immediately clear. It is possible that the characteristics of the intervention design led to superior uptake in some specialties. A recent systematic review of the impact of computerized clinical decision support found several characteristics of SUPPORT are associated with greater impact: automatic rather than elective appearance of the decision support tool, display on-screen rather than on paper, and inclusion of patient-specific suggestions [13]. It is possible that our design, in which the screening question response prompts the appearance of tailored guidance depending on the answer chosen, was particularly effective in motivating uptake of the screener. We did not collect data on provider attendance to roll-out meetings or document views of the brief video explainer of the SUPPORT build provided to all Fish Center clinicians. It is possible that the brief nature of our training—one video of less than 5 minutes and one meeting of less than 1 hour—enabled saturation of the

intervention in the target population despite their busy schedules, but we do not have process data test this hypothesis.

This study should be interpreted in light of several limitations. A large proportion (54.6%) of patients under 25 stated they do not want a pregnancy in the future, though at a population level 86% of AFAB individuals in the United States do give birth at some point in their lives [14]. This discrepant finding could indicate that the Fish Center patient population differs from the population overall, but likely reflects that pregnancy intentions are not stable across the life course, reinforcing the importance of repeated periodic screenings. SUPPORT was designed for use in EMR systems and replicating or implementing this build could be cost-prohibitive in health care systems without robust EMR infrastructure and support. In addition, this study was implemented in a practice with a clinical focus on the care of women and AFAB individuals, and clinicians in this practice may be more comfortable with or aware of the importance of pregnancy intention assessment compared to providers overall. While the intervention achieved high use of screening during the COVID-19 pandemic compared to similar prior interventions, likely uptake would have been even higher in different conditions. Additionally, our patient population lacked racial diversity, potentially limiting the generalizability of our results.

SUPPORT showed high promise as a systems-level intervention to encourage appropriate screening of pregnancy-capable individuals in primary care and disease specialty practices as well as in obstetrics and gynecology. Future studies should investigate the use of SUPPORT in a wider range of clinical contexts and explore whether screening rates are higher during less acute phases of the ongoing COVID-19 pandemic. Future studies with larger sample sizes can also track patient outcomes over time to assess whether screening is correlated with differences in pregnancy rates and pregnancy or disease outcomes. Because SUPPORT is non-proprietary, implementing this intervention at scale may be more feasible for health systems, particularly safety net providers, than other screeners which require the payment of a fee for use.

## Acknowledgments

The authors thank Kathy Kehoe and Harika Dabbara for practical support during intervention implementation, and Holly Barr Vermilya, Marianne Moore, and the Partners eCare Research team for implementing the custom SUPPORT infrastructure in the medical record.

## Author Contributions

**Conceptualization:** Elizabeth Janiak, Kathryn Rexrode, Maya Behn, Kari P. Braaten, Candace H. Feldman.

**Data curation:** Elizabeth Janiak, Leah Santacroce, Maya Behn.

**Formal analysis:** Elizabeth Janiak, Leah Santacroce, Sarah L. Johns, Candace H. Feldman.

**Funding acquisition:** Kathryn Rexrode.

**Investigation:** Elizabeth Janiak, Kathryn Rexrode, Kari P. Braaten, Candace H. Feldman.

**Methodology:** Elizabeth Janiak, Maya Behn, Kari P. Braaten, Candace H. Feldman.

**Project administration:** Sarah L. Johns, Maya Behn.

**Resources:** Maya Behn.

**Supervision:** Maya Behn.

Writing – original draft: Elizabeth Janiak, Leah Santacroce, Sarah L. Johns, Candace H. Feldman.

Writing – review & editing: Elizabeth Janiak, Kathryn Rexrode, Leah Santacroce, Sarah L. Johns, Maya Behn, Kari P. Braaten, Candace H. Feldman.

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
