## [Decision Letter · Decision Letter 0]

20 Dec 2023

PONE-D-23-28891Screening to Understand Pregnancy Preferences and Offer Referrals and Treatment (SUPPORT): Results of a Pilot Quality Improvement InitiativePLOS ONE

Dear Dr. Janiak,

Thank you for submitting your manuscript to PLOS ONE. After careful consideration, we feel that it has merit but does not fully meet PLOS ONE’s publication criteria as it currently stands. Therefore, we invite you to submit a revised version of the manuscript that addresses the points raised during the review process.

**ACADEMIC EDITOR COMMENT: **

Please ensure that figure 1 is attached to the manuscript

We look forward to receiving your revised manuscript.

Kind regards,

Maher Abdelraheim Titi

Academic Editor

PLOS ONE

“This research was supported by internal funding provided through the Division of Women’s Health at Brigham and Women’s Hospital.”

4. Please upload a copy of Figure 1, to which you refer in your text on page 6. If the figure is no longer to be included as part of the submission please remove all reference to it within the text.

5. Please include your tables as part of your main manuscript and remove the individual files. Please note that supplementary tables (should remain/ be uploaded) as separate "supporting information" files

Reviewers' comments:

Reviewer's Responses to Questions

**Comments to the Author**

1. Is the manuscript technically sound, and do the data support the conclusions?

Reviewer #1: Yes

Reviewer #2: Yes

2. Has the statistical analysis been performed appropriately and rigorously? 

Reviewer #1: Yes

Reviewer #2: Yes

3. Have the authors made all data underlying the findings in their manuscript fully available?

Reviewer #1: No

Reviewer #2: No

4. Is the manuscript presented in an intelligible fashion and written in standard English?

Reviewer #1: Yes

Reviewer #2: Yes

5. Review Comments to the Author

Reviewer #1: The manuscript reports on a pilot study to assess the feasibility of integrating pregnancy intention screening into routine care through electronic records prompts. The authors do a good job of explaining the context of the study and results. The study methods are appropriate for the research aim. I have a few suggestions to be considered.

The introduction goes into some detail about the importance of these questions for people with chronic diseases yet the rest of the paper including the title, abstract, aims and discussion do not mention this group being a focus of the study. Consider incorporating into the rest of the paper.

The setting of the study (that in a center for women’s health focusing on those with chronic conditions) could be better described in the abstract. Acronym AFAB is used in the abstract but not defined. The explanation that paused due to COVID is not really needed in abstract.

Did clinicians prescreen if people were not able to become pregnant (other than those stated – documented hysterectomy or other sterilizing procedure) and may have been sensitive to questions about pregnancy?

A patient demographics table or description is not included here but could help to understand the study population- including their age and race (as mentioned in limitations sections of discussion). It could also include % with a chronic condition if that information is available.

If available include the number of clinicians involved in the study

The paper would also benefit from some minor editorial changes

• Line 180 Include the number of screenings in the text eg OB/GYN (353 screenings).

• Line 197 – change (ref) to (reference group)

• Line 189-192. Consider moving this sentence to start of next paragraph to keep the topic of contraception together.

• Ensure using consistent terms in text and Tables. Primary care is used in text but Internal medicine used in Table 1.

• Since using aOR in the results text, use adjusted odds ratio in table 2 with a footnote that model is adjusted by linear age.

Reviewer #2: This study makes a helpful contribution to the literature on screening for reproductive desires in clinical practice in the United States. I have a few suggestions for clarity and accuracy within the text:

1) In the abstract, spell out the meaning of AFAB before using the acronym.

2) Line 66 says “women and pregnancy-capable individuals,” but I think all of the studies you cite (references 2-4) say they studied women or “adult females” without identifying pregnancy capability. I appreciate your effort to be gender inclusive, but I think in citing other studies it’s best to carry forward their language, and then you can make a separate note that it’s unknown whether people of all genders capable of pregnancy were included, or it’s unfortunate that they weren’t. (If you disagree and find that, for example, transgender men capable of pregnancy were included in these studies, then please ignore my comment.) In your sentence, “women with chronic diseases” would also be simpler/easier for the reader to follow.

3) Line 74-75, for reader clarity, I think it would help to name/define the tools (PATH, OKQ) first and then briefly review the literature on them. As written, as each tool arises (e.g. PATH) the reader may not know what this is without prior introduction.

4) Line 111, “wide variety of clinical specializations and subspecialty clinics” sounds redundant

5) Line 139, “on site, at-the-elbow coaching” also sounds redundant

6) Line 145 says one utility measurement is “likelihood of contraceptive documentation.” Can you define this more? Was this based on asking the patient? Or on med list, problem list, a checkbox, or other field? What contraceptive methods does it include (e.g. are withdrawal, fertility awareness, or abstinence included)?

7) Line 194 says “using” contraception, but this seems different from the documentation outcome you’re measuring. (Unless the documentation is specifically of current contraceptive use and you feel this is a reliable indicator of patient behavior.)

8) Lines 218-220 comparing your findings to other studies: it would be helpful to write out in more detail what those other studies observed.

9) Line 247, in the Discussion you say your population lacked racial diversity, but you haven’t shared anything about the racial identities of the population.

6. PLOS authors have the option to publish the peer review history of their article (what does this mean?). If published, this will include your full peer review and any attached files.

Reviewer #1: No

Reviewer #2: No

---

## [Author Response · Author response to Decision Letter 0]

13 Mar 2024

Please see the Response to Reviewers document attached to our resubmission.

Please also note that we have updated our statement on data sharing and have fixed the author affiliation error noted.

---

## [Decision Letter · Decision Letter 1]

3 May 2024

Screening to Understand Pregnancy Preferences and Offer Referrals and Treatment (SUPPORT): Results of a Pilot Quality Improvement Initiative

PONE-D-23-28891R1

Dear Dr. Janiak,

We’re pleased to inform you that your manuscript has been judged scientifically suitable for publication and will be formally accepted for publication once it meets all outstanding technical requirements.

Kind regards,

Maher Abdelraheim Titi

Academic Editor

PLOS ONE

Additional Editor Comments :

All of the both reviewer’s comments have been addressed effectively, and the manuscript has significantly improved. I believe that the manuscript is now suitable for publication in PLOS ONE.

Reviewers' comments:

Reviewer's Responses to Questions

**Comments to the Author**

1. If the authors have adequately addressed your comments raised in a previous round of review and you feel that this manuscript is now acceptable for publication, you may indicate that here to bypass the “Comments to the Author” section, enter your conflict of interest statement in the “Confidential to Editor” section, and submit your "Accept" recommendation.

Reviewer #1: All comments have been addressed

2. Is the manuscript technically sound, and do the data support the conclusions?

Reviewer #1: Yes

3. Has the statistical analysis been performed appropriately and rigorously? 

Reviewer #1: Yes

4. Have the authors made all data underlying the findings in their manuscript fully available?

Reviewer #1: Yes

5. Is the manuscript presented in an intelligible fashion and written in standard English?

Reviewer #1: Yes

6. Review Comments to the Author

Reviewer #1: (No Response)

7. PLOS authors have the option to publish the peer review history of their article (what does this mean?). If published, this will include your full peer review and any attached files.

Reviewer #1: No

---

## [Editor Report · Acceptance letter]

19 Jul 2024

PONE-D-23-28891R1 

PLOS ONE

Dear Dr. Janiak, 

I'm pleased to inform you that your manuscript has been deemed suitable for publication in PLOS ONE. Congratulations! Your manuscript is now being handed over to our production team.

Kind regards, 

on behalf of

Dr. Maher Abdelraheim Titi 

Academic Editor

PLOS ONE